# Numerical Simulation of Taylor—Couette—Poiseuille Flow at Re = 10,000

Andrey Gavrilov [1,†] and Yaroslav Ignatenko [2,*,†]

1 Laboratory of Physical and Chemical Technologies for the Development of Hard-to-Recover Hydrocarbon Reserves, Siberian Federal University, Svobodny Ave., 82A, 660041 Krasnoyarsk, Russia; gavand@yandex.ru
2 Baker Hughes, Baker-Hughes-Straße 1, 29221 Celle, Germany
* Correspondence: yaroslav.ignatenko@gmail.com; Tel.: +49-152-314-2-36-46
† These authors contributed equally to this work.

**Abstract:** A fully developed turbulent flow in a concentric annulus, Re $= 10,000$, $r_i/r_o = 0.5$, with an inner rotating cylinder in the velocity range $N = U_\omega/U_b = 0 \div 4$, is studied via a large-eddy simulation. Also, for comparison, simulations by steady-state, unstationuary RANS $k$-$\omega$ SST (URANS), and Elliptic Blending Model (EBM) were made. The main focus of this study is on the effect of high rotation on the mean flow, turbulence statistics, and vortex structure. Distribution of the tangential velocity and the Reynolds stress tensor change their behaviour at $N > 0.5 \sim 1$. With rotation increases, the production of tangential fluctuation becomes dominant over axial ones and the position of turbulent kinetic energy maximum shifts towards the wall into the buffer zone. URANS and EBM approaches show good agreement with LES in mean flow, turbulent statistics, and integral parameters. The difference in pressure loss prediction between LES and URANS does not exceed 20%, but the average difference is about 11%. The EBM approach underestimates pressure losses up to 9% and on average not more than 5%. Vortex structures are described well by URANS.

**Keywords:** concentric annulus; turbulent rotating flow; rotating inner cylinder; large-eddy simulation; URANS; elliptic blending model

## 1. Introduction

The annular flow phenomenon is ubiquitous in engineering and can be observed in diverse applications such as heat exchangers, chemical mixing devices, sliding bearings, well drilling, and turbomachinery. As a result, numerous experimental and numerical studies have been conducted to investigate the flow characteristics associated with the internal and external rotation of the cylinder (wall). Despite the geometric simplicity of this configuration, annular flow exhibits a complex three-dimensional boundary layer, swirling, and turbulence enhancement when the inner wall rotates. Conversely, when the outer wall rotates, the flow is suppressed, leading to a tendency toward laminarization [1].

Most of the previous research has focused on the inner wall rotational flow, apparently because of the interest in the shear-driven three-dimensional boundary layer. Previous experiments conducted by Nouri et al. (1993–1997) [2–4], Escudier and Gouldson (1995) [5], and Rothe and Pfitzer (1997) [6] have provided valuable insights into the influence of inner wall rotation on turbulence structures in the inner wall region. A DNS study by Chung et al. (2002) [7], while limited to a non-rotating annular flow at a Reynolds number of 8900, discussed the effect of convex and concave lateral curvature on turbulence dynamics in regions adjacent to the inner and outer wall, respectively. Another DNS investigation conducted by Jung and Sung (2006) [8] used DNS to explore coherent structures near the inner rotating wall and their modification due to the work of centrifugal forces. In contrast, investigations of annular flows with a rotating outer wall are relatively scarce.

In the literature, a common observation is that the rotation rate in annular flows is usually low to moderate, where the velocity of the moving wall is either slightly above

or equal to the fluid bulk velocity. For instance, Chung and Sung (2005) [9] conducted LES of annular flow at Re $= 8900$ and investigated three rotation rates ($N = 0.2145$, $N = 0.429$, $N = 0.858$), which exhibited discernible but not significant variations for different values of $N$.

Annular flow in the narrow gap was investigated by LES in studies [10–12]. The authors [10] investigated the influence of the rotation on the mean flow and on the turbulence statistics, and studied the nature of the coherent structures appearing in the boundary layer and their influences on the heat transfers. Authors [12] showed a large contribution of the turbulent transport term on the Nusselt number results. In article [11], the turbulent kinetic energy transport in the near wall region was investigated for heat recovery system application.

Hanjalić and Launder in book [13] note that RANS models using the eddy viscosity hypothesis cannot properly describe flows with a high degree of swirl. This was also shown in a list of publications [14–17] about the Taylor–Couette–Poiseuille annular flow.

Authors of [16] compare results of LES and URANS approaches for Newtonian and power-law fluid. They prove and explain the reason for the applicability of the URANS approach to simulate swirling flow in the annulus.

In study [14], authors show good agreement with the first and second moment of flow simulated by RANS RSM [18] with the experimental data of [5]. Applied model [18] is inspired by studies [19–21].

Here in the introduction, we mentioned only key studies related to current work. The work of Lockett should also be noted [22] for the extensive review of experimental and numerical work on the subject done before the 1990s. An extensive and fresh literature review of heat transport in rotating annular ducts was done in [23]. The review includes articles from the first work by Taylor to recent experimental and numerical studies. A wide range of numerical methods was observed. There are not only articles related to heat transfer but also about isothermal annulus flow.

As mentioned above in the literature review, the Taylor–Couette–Poiseuille system has been extensively investigated for $N < 1$. One of the purposes of the present study is to extend the investigated range of the Reynolds number to Re$= 10,000$ and the rotation number up to $N = 4$ using LES. Another goal of this work is to compare LES results with results obtained by URANS $k$-$\omega$ SST, RANS $k$-$\omega$ SST, and RANS EBM [24] approaches. The first and second flow moments obtained by the EBM approach and URANS were compared with LES. Integral parameters such as pressure losses and torque were compared for all above-mentioned approaches.

## 2. Problem Statement and Numerical Algorithm

Consider the incompressible flow throw an annular axisymmetric channel. The cylinder axis is aligned with the z-axis of the coordinate system. The inner cylinder, with radius $r_i$, rotates clockwise (viewed toward z-direction) with constant velocity $U_\omega$, while the outer cylinder, with radius $r_o$, is at rest. The ratio of the inner to outer cylinder radius is $r_i/r_o = 0.5$. The Reynolds number Re$= 10,000$ is based on fluid bulk velocity $U_b$ and hydraulic diameter $D_h = 2(r_o - r_i)$. The dimensionless rotation velocity of the inner cylinder takes values $N = U_\omega/U_b = 0, 0.2, 0.5, 1, 2$ and $4$.

The simulation is carried out using an in-house CFD code, which has been verified by DNS of the flow in the pipe [25] and by LES in annulus [16]. The numerical algorithm is based on the finite volume method for an unstructured mesh. The SIMPLE–C algorithm was applied for the pressure correction procedure and the collocated grid arrangement with the Rhie–Chow interpolation. The system of linear algebraic equations for the pressure correction equation is solved using an algebraic multigrid solver. Periodic boundary conditions are applied for the flow direction.

In the LES approach, here the filtered Navier–Stokes equations and the continuity equation were closed by the dynamic Germano–Lilly eddy viscosity sub-grid model [26,27]. The convective and diffusive terms were approximated by a second-order central difference

scheme. The time step discretization was carried out by the second-order Crank–Nicholson scheme. The cell size on the channel walls in wall units was chosen so that in the radial direction $\Delta r^+ < 1$, in the axial direction $\Delta z^+ < 16$, and in the tangential direction $(\Delta \theta r)^+ < 20$. The sizes of the wall cells were chosen on the basis of recommendations from the book [28] and previous works on the same subject [1,9,11]. From the walls of the computational domain to the middle of the gap, the mesh sizes were increased with a growth factor of no more than 5% from mesh to mesh. Thus, the mesh is redundant for LES, since the modelled part of TKE did not exceed 9% in most cases (Figure A1), with recommendations up to 20% [29]. The time step was chosen based on the condition CFL < 0.9. Discussion about choosing of the time step and meshing is placed in Appendix A. The channel length $L_z = 4.5 D_h$ was chosen based on the results of [8], where a two-point correlation was used to determine the sufficiency of the channel length. The parameters of the computational mesh for LES are shown in Table 1.

**Table 1.** Mesh resolution applied at LES for different rotation rates N.

| $N$ | 0 | 0.2 | 0.5 | 1 | 2 | 4 |
|---|---|---|---|---|---|---|
| $N_r \times N_\theta \times N_z$ | $103 \times 129 \times 120$ | $103 \times 515 \times 403$ | $119 \times 473 \times 376$ | $109 \times 273 \times 342$ | $117 \times 421 \times 659$ | $141 \times 470 \times 480$ |
| $\Delta r_i^+$ | $0.85 \pm 0.14$ | $0.79 \pm 0.14$ | $0.79 \pm 0.16$ | $0.98 \pm 0.21$ | $0.93 \pm 0.2$ | $0.55 \pm 0.25$ |
| $\Delta r_o^+$ | $0.74 \pm 0.13$ | $0.67 \pm 0.12$ | $0.47 \pm 0.08$ | $0.71 \pm 0.13$ | $0.78 \pm 0.15$ | $0.75 \pm 0.12$ |
| $\Delta z_i^+$ | $8.22 \pm 1.37$ | $7.63 \pm 1.35$ | $9.56 \pm 1.91$ | $13.03 \pm 2.83$ | $9.36 \pm 2.05$ | $14.47 \pm 3.19$ |
| $\Delta z_o^+$ | $7.68 \pm 1.31$ | $6.97 \pm 1.22$ | $7.79 \pm 1.37$ | $9.44 \pm 1.75$ | $5.9 \pm 1.14$ | $16.11 \pm 2.25$ |
| $(\Delta \theta r)_i^+$ | $4.49 \pm 0.75$ | $4.18 \pm 0.74$ | $5.31 \pm 1.06$ | $11.41 \pm 2.48$ | $10.24 \pm 2.24$ | $20.24 \pm 4.46$ |
| $(\Delta \theta r)_i^+$ | $8.37 \pm 1.43$ | $7.61 \pm 1.33$ | $8.65 \pm 1.52$ | $16.51 \pm 3.05$ | $12.89 \pm 2.49$ | $11.28 \pm 2.25$ |
| CFL | $0.67 \pm 0.28$ | $0.75 \pm 0.29$ | $0.73 \pm 0.25$ | $0.7 \pm 0.21$ | $0.73 \pm 0.23$ | $0.88 \pm 0.27$ |

Other non-stationary simulations were carried out using URANS with the $k$-$\omega$ SST turbulence model. The numerical discretization schemes used are the same as in the LES approach and have the same radial distribution of computational nodes. In the tangential and axial directions, the number of nodes is reduced but satisfied $\Delta z^+ < 30$, $(\Delta \theta r)^+ < 30$. The time step is also chosen based on the condition CFL< 0.8 and channel length $L_z = 4.5 D_h$. The parameters of the computational mesh are shown in Table 2.

**Table 2.** Mesh resolution applied at URANS for different rotation rates N.

| $N$ | 0.2 | 0.5 | 1 | 2 | 4 |
|---|---|---|---|---|---|
| $N_r \times N_\theta \times N_z$ | $103 \times 129 \times 120$ | $119 \times 138 \times 120$ | $109 \times 148 \times 144$ | $117 \times 180 \times 216$ | $132 \times 234 \times 312$ |
| $\Delta r_i^+$ | $0.8 \pm 0$ | $0.82 \pm 0.14$ | $0.94 \pm 0.2$ | $0.9 \pm 0.22$ | $0.74 \pm 0.16$ |
| $\Delta r_o^+$ | $1.42 \pm 0$ | $1.2 \pm 0.02$ | $1.2 \pm 0.12$ | $1.0 \pm 0.1$ | $0.8 \pm 0.06$ |
| $\Delta z_i^+$ | $25.61 \pm 0$ | $29.32 \pm 4.97$ | $29.39 \pm 6.53$ | $27.58 \pm 6.53$ | $28.55 \pm 6.41$ |
| $\Delta z_o^+$ | $24.06 \pm 0$ | $24.09 \pm 0.5$ | $21.32 \pm 1.93$ | $16.97 \pm 1.63$ | $15.69 \pm 1.35$ |
| $(\Delta \theta r_i)^+$ | $16.64 \pm 0$ | $17.94 \pm 3.04$ | $20.11 \pm 4.47$ | $23.24 \pm 5.5$ | $26.7 \pm 6$ |
| $(\Delta \theta r_o)^+$ | $31.22 \pm 0$ | $29.43 \pm 0.61$ | $29.13 \pm 2.63$ | $28.57 \pm 2.74$ | $29.32 \pm 2.52$ |
| CFL | $0.44 \pm 0.17$ | $0.46 \pm 0.16$ | $0.60 \pm 0.18$ | $0.77 \pm 0.29$ | $0.63 \pm 0.19$ |

In the LES and URANS approaches, all statics were obtained by averaging over time and in uniform directions, along the channel, and in the angular direction. The averaging time in all transient simulations was about $200 D_h / U_b$.

For steady-state simulations, two RANS approaches were also used for comparison. The first one is widely used in engineering applications: the $k$-$\omega$ SST with eddy viscosity assumption. The second approach, based on the Reynolds Stress Model (RSM), which models the full Reynolds stress tensor, is Elliptic Blending Model (EBM) [24]. For both of these approaches, mesh with two cells in an axial direction was used, and the same radial distribution of cells was used as in the LES approach. The convective terms were approximated by a QUICK scheme.

### 3. Results and Discussion

#### 3.1. Mean Flow Properties and Turbulence Statistics

It should be noted that the flow features and redistribution mechanics of the turbulence characteristics have been sufficiently described in earlier works. In this section, we focus on flow features that have not been noted before and compare the LES results with the results of the EBM simulation.

An increase in rotation increases the axial velocity gradient on the walls, making the axial velocity distribution $U_z$ flatter in the central region of the gap. The EBM model slightly overestimates the axial velocity near walls (Figure 1a), thereby underestimating it in the central region. The azimuthal velocity component $U_\theta$ at $N = 0.2$–$0.5$ rotations has small local maxima at a distance 0.2–0.4 of the gap from the inner wall(Figure 1b). The EBM does not capture this small effect but tends to the LES curves at larger rotations.

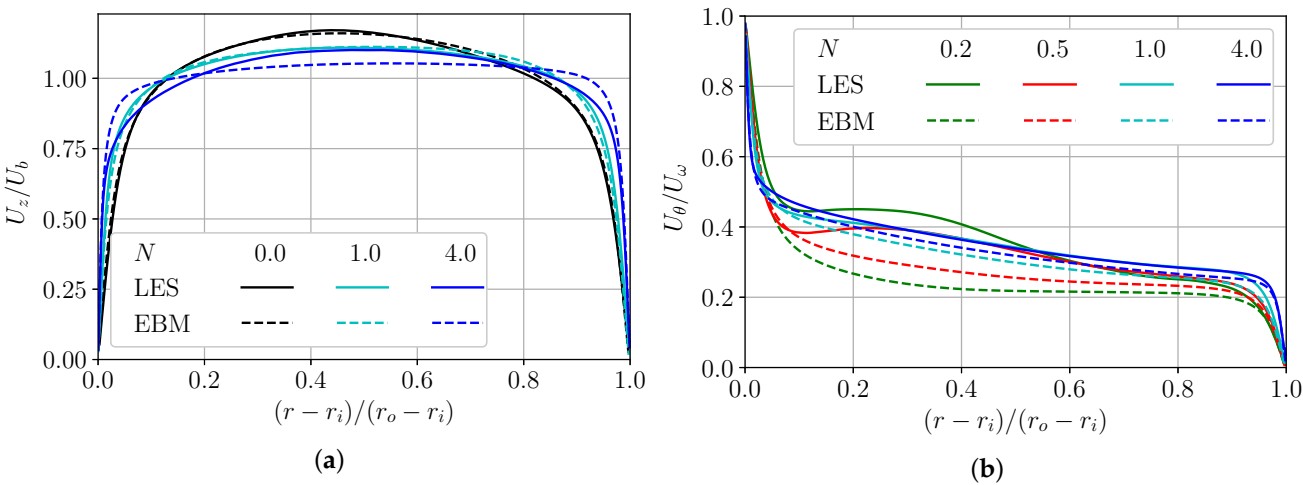

**(a)**  **(b)**

**Figure 1.** Axial (**a**) and tangential (**b**) velocity normalized by the bulk and inner cylinder velocity, correspondingly. Solid curves show LES and the dashed ones shows EBM.

Let us discuss the axial velocity profile of the inner and outer walls in wall unites. In the non-rotating case, the axial velocity profile fits well to the wall–law near the inner and outer walls of the channel. As in work [9], the inner wall axial velocity profile $U_{z,i}$ appreciably decreases with increasing rotation $N$ (Figure 2a). This is explained by the increasing in the axial friction on the inner wall with $N$. However, near the outer wall, these differences are noticeably smaller (Figure 2b). It is also remarkable that with rotation of $N = 0.2$, the axial velocity is higher than in the case without rotation, but with increasing rotation, as near the inner wall, it decreases noticeably in the buffer and logarithmic zone. The EBM approach does not capture this small effect at 0.2 rotation, but the general trend in behaviour and numerical values are captured correctly.

Wide theoretical investigation of swirling turbulent flow in axially symmetric flow was made in [30]. Here, we should shortly repeat some logic and conclusions to describe the results of the simulation. Let us briefly overview relations between different terms in axially symmetric systems with centrifugal force. The swirl effect on turbulent flow is determined by $\partial(rU_\theta)/\partial r$ and $\overline{u_r u_\theta}$. The classification made in the three cases according to the signs of these two terms are

- Case 1: $\overline{u_r u_\theta} \leq 0$ and $\partial(rU_\theta)/\partial r > 0$.
- Case 2: $\overline{u_r u_\theta} > 0$ and $\partial(rU_\theta)/\partial r \leq 0$.
- Case 3: $\overline{u_r u_\theta} > 0$ and $\partial(rU_\theta)/\partial r > 0$.

The first case realizes when the outer cylinder rotates; this suppresses turbulence and leads to flow laminarization, which is shown in the study [1]. When only the inner cylinder rotates, the $\overline{u_r u_\theta}$ value is positive along the entire annular gap, but the value of $\partial(rU_\theta)/\partial r$ can change sign, thus, here we should consider the next two cases.

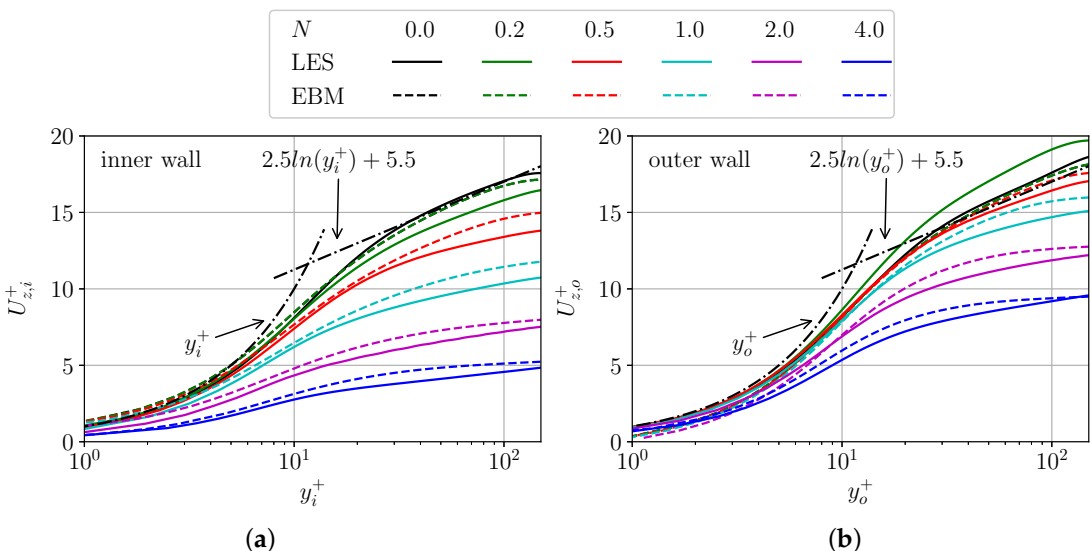

**Figure 2.** Axial velocity in wall units. Inner wall region (**a**) and outer wall region (**b**). Solid curves show LES and the dashed curved shows EBM.

The mechanism of the rotation effect on momentum transport is not straightforward; therefore, we show it using a block diagram for Case 2 and Case 3 in Figure 3, which is inspired by [30]. Expressions for the production of the various components of the stress tensor are given below:

$$P_{zz} = -2\overline{u_z u_r}\frac{\partial U_z}{\partial r}, P_{rr} = 2\overline{u_r u_\theta}\frac{U_\theta}{r}, P_{\theta\theta} = -2\overline{u_r u_\theta}\frac{\partial U_\theta}{\partial r}, \tag{1}$$

$$P_{zr} = \underbrace{-\overline{u_r u_r}\frac{\partial U_z}{\partial r}}_{(P_{zr})_I} + \underbrace{\overline{u_\theta u_z}\frac{U_\theta}{r}}_{(P_{zr})_{II}}, P_{\theta z} = \underbrace{-\overline{u_r u_\theta}\frac{\partial U_z}{\partial r}}_{(P_{\theta z})_I} - \underbrace{\frac{\overline{u_z u_r}}{r}\frac{\partial r U_\theta}{\partial r}}_{(P_{\theta z})_{II}}, P_{r\theta} = -\overline{u_r u_r}\frac{\partial U_\theta}{\partial r} + \overline{u_\theta u_\theta}\frac{U_\theta}{r}. \tag{2}$$

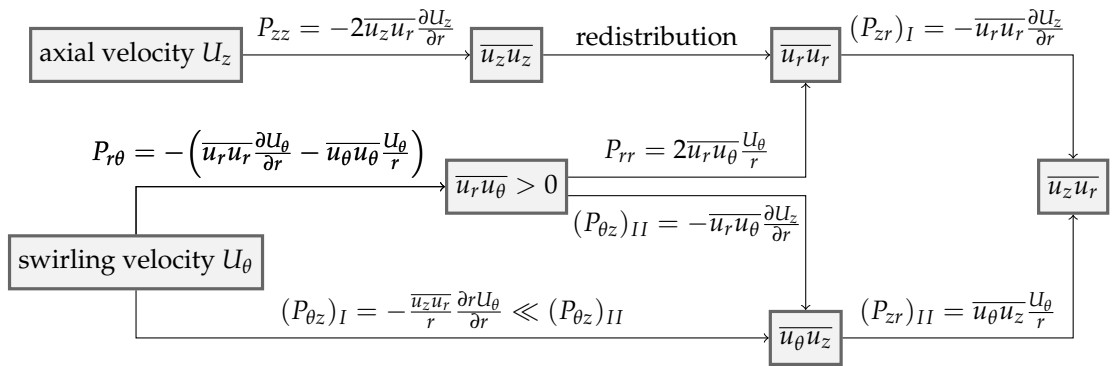

**Figure 3.** The mechanism of swirl effects on momentum transport for Case 2 and Case 3.

The axial fluctuations $\overline{u_z u_z}$ promoted by axial flow gradient $\frac{\partial U_z}{\partial r}$ with $\overline{u_z u_r}$ because they change sign at the same time as the walls, making production positive ($P_{zz} = -2\overline{u_z u_r}\frac{\partial U_z}{\partial r}$). Through redistribution, it promotes the radial velocity fluctuations $\overline{u_r u_r}$, which also are increased due to their positive production term $P_{rr} = 2\overline{u_r u_\theta}U_\theta/r$.

Stress $\overline{u_r u_\theta} > 0$ that is responsible for tangential shear stress and torque is produced by rotation. That stress $\overline{u_r u_\theta}$ is part of the production of radial velocity fluctuation $\overline{u_r u_r}$ and stress $\overline{u_\theta u_z}$. Stress $\overline{u_\theta u_z}$ is generated by the two production terms $-\overline{u_r u_\theta}\partial U_z/\partial r$ and $-(\overline{u_z u_r}/r)\partial(rU_\theta)/\partial r$. In the current simulations, term $(P_{\theta z})_I = -(\overline{u_z u_r})/r\partial(rU_\theta)/\partial r$ changes sign along the radius, but at the same time is significantly smaller than the value of $(P_{\theta z})_I = -\overline{u_r u_\theta}\partial U_z/\partial r$. By these means, Case 2 and Case 3 in the present conditions



give the same results, and term $\overline{u_\theta u_z}$ is promoted. Thus, the generation term of $\overline{u_z u_r}$ $((P_{zz})_{II} = 2\overline{u_\theta u_z}U_z/r)$ is also enhanced. Stress $\overline{u_r u_\theta}$ is also enhanced by the increase in $2\overline{u_\theta u_z}U_\theta/r$.

The term responsible for the axial momentum transfer from the walls $\overline{u_z u_r}$ is generated by the two production terms $-\overline{u_r u_r}\frac{\partial U_z}{\partial r}$ and $\overline{u_\theta u_z}\frac{U_\theta}{r}$. The first one depends on the gradient of the axial velocity and radial fluctuations promoted by the axial and rotational velocity. The second term is promoted by rotation. Thus, the term $\overline{u_z u_r}$ responsible for axial friction (pressure loss) increases with both axial speed and rotational velocity.

The distribution of root-mean-square velocity fluctuations and Reynolds shear stresses obtained by LES and EBM are shown in Figure 4a. Without rotation, the axial component of the velocity fluctuations dominates over the other components and has well-defined maximums near the channel walls, similar to the flow in the channel. Rotation leads to an increase in all velocity fluctuation components as centrifugal force leads to flow destabilization and greater turbulence; the angular component appears with maximums near the channel walls like the axial component, and at high values of rotation their maximums become comparable in magnitude. It should also be noted that the position of the fluctuation peak gets closer to the wall with rotation as the viscous sublayer becomes thinner. The EBM approach describes the behavior of velocity fluctuation curves well, except for the axial fluctuations $u_{z,rms}$, underestimating it. Thus, the total kinetic energy of velocity fluctuations is underestimated by EBM.

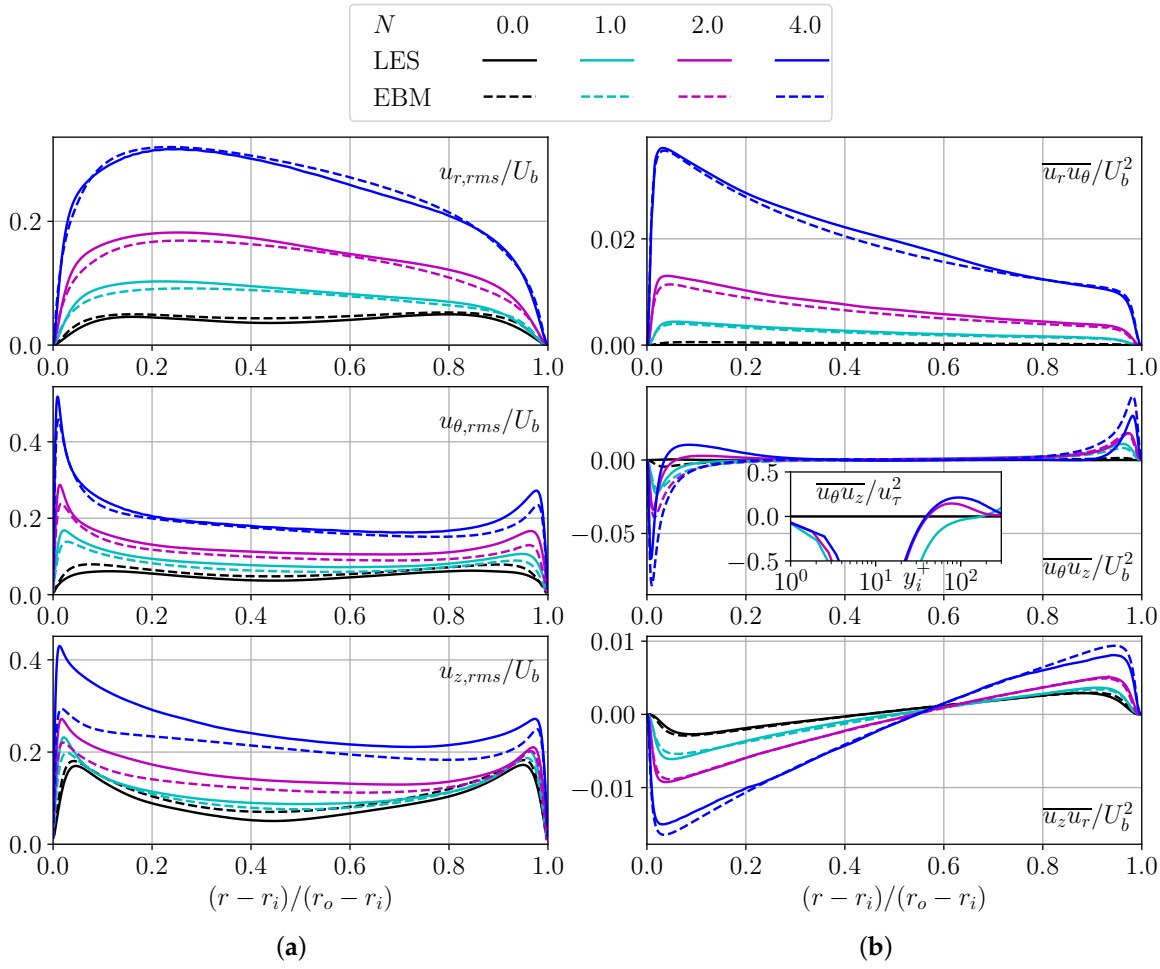

**Figure 4.** Distributions of root-mean-square velocity fluctuations (**a**) and Reynolds shear stress (**b**). Solid curves show LES, and the dashed one shows EBM.

At the transition from $N = 1$ to $N = 2$, the stress shape $\overline{u_\theta u_z}$ changes behavior along the radius. At the region of the inner wall and $N < 2$, stress $\overline{u_\theta u_z}$ is strictly negative, but at

$N \geq 2$ it changes sign and has a local maximum (Figure 4b). The mechanism of production and suppression of Reynolds stress tensor is well-described in article [30]. Production of $\overline{u_\theta u_z}$ consist of two terms, $P_{\theta z} = -\frac{\partial U_z}{\partial r}\overline{u_r u_\theta} - \frac{\partial U_\theta}{\partial r}\frac{\overline{u_z u_r}}{r}$. The second term changes the sign, but it is too small to change the sign of production. It would be logical to assume that the redistribution term is responsible for the behavioural change of the component $\overline{u_\theta u_z}$. EBM misses such a small effect; below, we will see that this is not reflected in the integral characteristics.

Production terms $P_{zz}$ and $P_{\theta\theta}$ have same structure (Formula (1)) and they depend on stress responsible for the transfer of respective momentum in radial direction and respective velocity gradient. It is evident that the tangential gradient $\partial U_\theta / \partial r$ grows faster than the axial gradient $\partial U_z / \partial r$, with increase in rotation $N$ at inner wall. At the same time, rotation leads to an increase in wall friction and friction velocity $U_\tau$. Thus, production $P_{\theta\theta}$ of the rotational component increases, but production $P_{zz}$ of axial component decreases near the inner wall in wall units. Hence, the fluctuations of the rotational velocity component grow faster with rotation (Figure 4a). It is also worth noting that the maximum production shifts from the buffer zone closer to the viscous sublayer, probably due to the rotation of the inner cylinder and centrifugal force (Figure 5). The sum of productions $P_{zz}$ and $P_{\theta\theta}$ remains approximately constant near the rotating wall in near-wall units ($P_{rr}$ significantly smaller than both of them).

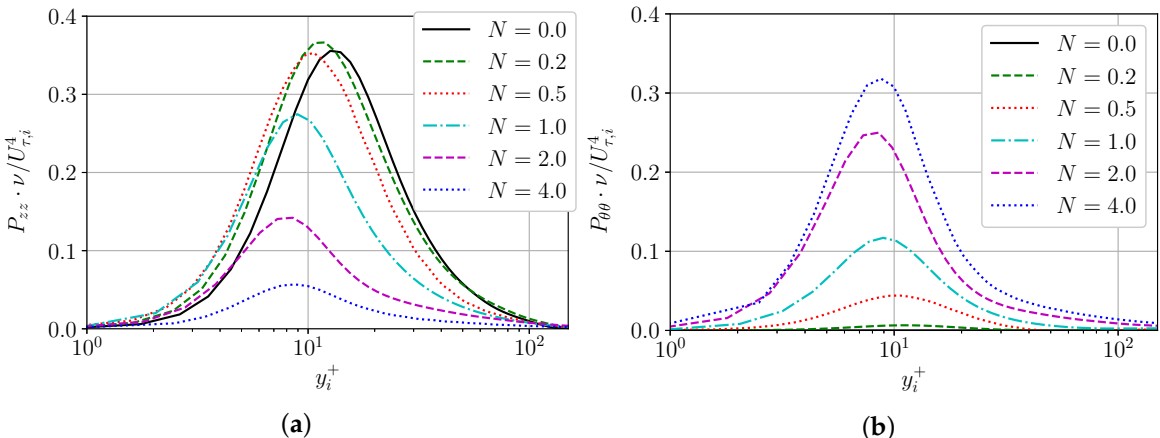

(**a**)   (**b**)

**Figure 5.** Production of $\overline{u_z u_z}$ (**a**) and $\overline{u_\theta u_\theta}$ (**b**) in wall units near the inner wall.

A comparison of TKE distribution obtained by LES, URANS, and EBM is shown in Figure 6. As mentioned above, the EBM underestimates the fluctuations of the axial velocity, hence the total kinetic energy of the fluctuations, but correctly captures the position of its maximums.

It was shown earlier that $k$-$\omega$ SST as an eddy viscosity model does not work properly in rotational flow, underestimating TKE. Thus, we do not show curves for it in Figure 6a. Let us try to describe why the URANS approach improves simulation results. Due to fluid ejection from the inner wall of a fluid layer with a low level of modelled turbulent energy, the thickness of the near-wall layer increases, shear stresses on the walls decrease, and the modelled turbulent energy dissipates. A decrease in the modelled turbulent viscosity leads to flow instability and formation of vortices in the near-wall region. The ejection process is responsible for suppression of the modelled part of the turbulence and an increase in the resolvable part of the turbulence caused by large-scale spiral vortices. Thus, the mechanism for increasing the resolvable fluctuation fraction is proportional to the rotation; this can be seen from Figure 6a. The URANS approach applied here reproduces values and a maximum of TKE well near the inner wall, but does not catch in at the outer wall of the annular channel. Thus, we can conclude that URANS probably resolves vortex structures at passive wall worth than at a wall that actively produces vortices by rotation. Also, we should note that URANS at rotation $N = 0$–0.2 does not resolve any vortices,

simulating turbulent steady-state flow, i.e., the resolvable TKE fraction is zero (Figure 6b). At rotation $N = 0.5$, vortex structures are resolved only near the inner cylinder by URANS. The resolved fraction of the TKE value tends to unity near the wall, the location of flow instability generation, and noticeably decreases towards the center of the gap. Interesting to note that the total resolved TKE increases at $N = 0$–1 and decreases at $N > 1$ (Figure 6b).

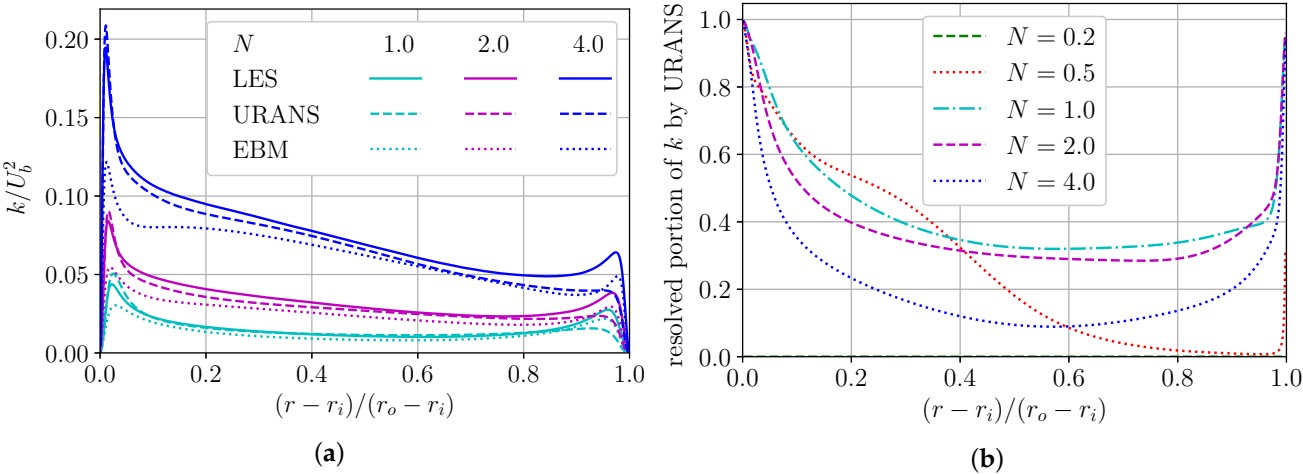

**Figure 6.** (**a**) Turbulent kinetic energy $k$ (TKE) distribution. (**b**) Part of resolved TKE by URANS.

In Chang's work [9], it was shown that as $N$ grows, the structural parameter $a_1$ also grows, which indicates an increase in shear stresses relative to normal stresses, i.e., the efficiency of shear stress generation grows. In our simulations, this conclusion is confirmed for $N < 1$, but at larger rotations, the parameter $a_1$ decreases (Figure 7a). EBM cannot fully replicate the LES curves for parameter $a_1$, but the general character of the dependence is captured, which is a very good result for turbulent flow with swirling flow (Figure 7b). Also, it should be noted that EBM is not sensitive to small rotations, because curves for $N = 0$ and $N = 0.2$ are the same for EBM.

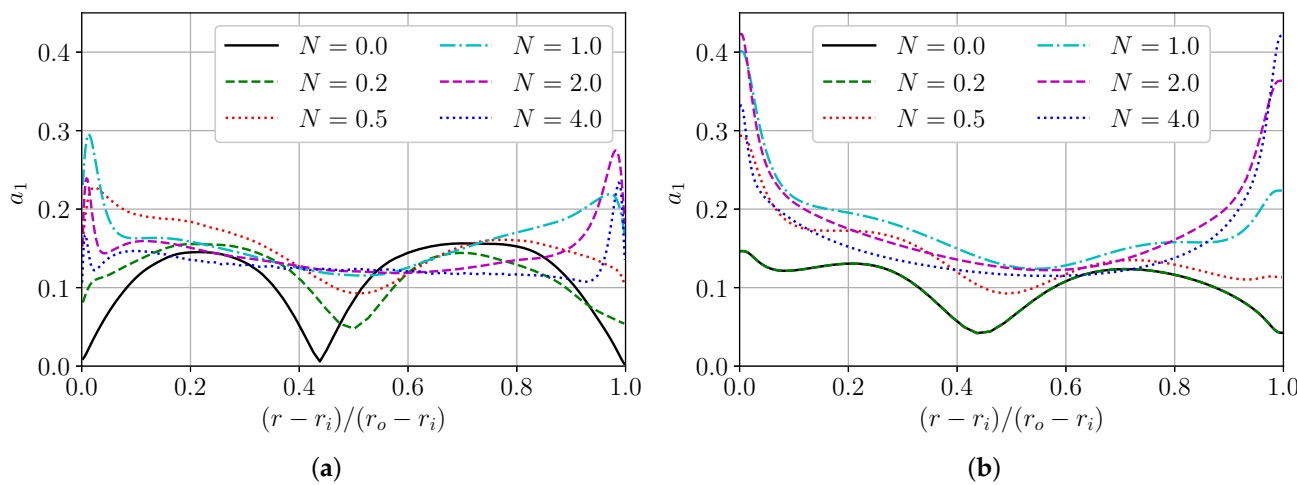

**Figure 7.** Structure parameter $a_1$. (**a**) LES and (**b**) EBM.

### 3.2. Flow Patterns, Vortical and Turbulence Structures

Briefly, flow visualization by $\lambda_2$ criterion were shown in [9] and [8] for Re = 8900, $r_i/r_o = 0.5$, and $N = 0, 0.429$. A more detailed flow structures analysis was made by [11] for the radii ratio $r_i/r_o = 0.809$ at the maximum Reynolds number Re = 8776 with rotation $N = 0.85$. Here, we concentrate our attention on high rotation and comparison of flow structures obtained by LES and URANS.

Classical Görtler instability was described when the boundary layer thickness is comparable to the radius of the curvature; under the action of centrifugal force, a pressure change in the boundary layer occurs [31]. In our case, the rotation of the inner cylinder generates a centrifugal force sufficient to destabilise the flow. Centrifugal instability of the boundary layer leads to the subsequent formation of Görtler vortices. The instability leads to the ejection of fluid from the channel walls, forming a pair of vortices and a region of reduced friction on the wall between them. A region of increased friction is formed between two pairs of such vortices, where fluid injects back. Schematically, ejection/injection, zone of higher/lower friction, and vortex formation are shown in Figure 8a. Visualisation of such vortices are shown in Figure 8b.

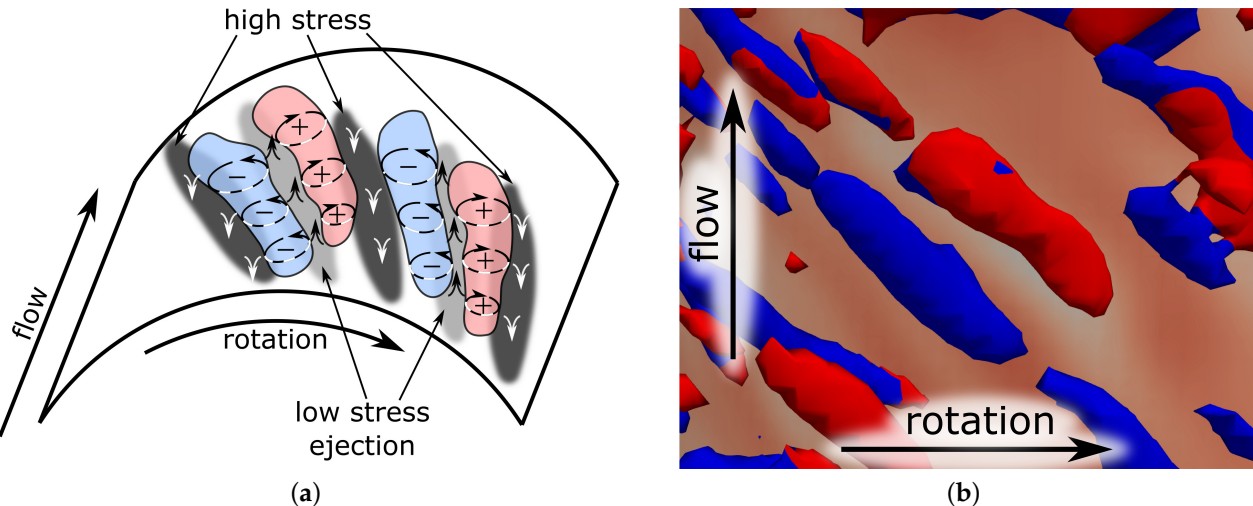

(**a**)  (**b**)

**Figure 8.** Vorticity formation on inner wall, their structure, and axial wall stress. (**a**) Schematic. (**b**) Flow visualization by $\lambda_2$ at $N = 2$. Red-coloured vortices have a positive rotation, blue ones have a negative rotation. Greater axial stresses on the wall are shown in a darker tone.

The velocity fluctuations is higher on the inner wall of the channel (Figure 6a), as it is an active generator of turbulence due to its rotation, while vortex formation on the outer wall is secondary, as it occurs due to rotation of the whole liquid near the resting wall. As a consequence, we should expect a higher density of vortices on the inner wall, as we can see from the LES and URANS visualizations (Figure 9a). The vortices are also distributed non-uniformly along the walls, forming helical clouds or clusters on both the inside and outside walls of the channel.

Vortices on different walls of the channel are inclined in opposite directions. On the inner cylinder, the spiral of the vortices is wrapped against the rotation of the flow and on the outer cylinder in the direction of the flow rotation. The eddies form in pairs and are carried downstream. As the inner wall is rotating, i.e., moving in a tangential direction relative to the liquid, the vortices formed will be carried away in the opposite direction to its movement. Thus, the direction of the vortices and the twisting of their spiral clusters is opposite to the rotation of the flow. In the case of the outer wall, the situation is exactly the opposite. Non-uniform distribution of vortices along the channel was observed in the Couette–Taylor flow in [32], where only the inner cylinder rotated without axial flow. If, in the case of purely rotational flow, the vortex clusters resemble a torus in case of a combination of rotational flow and axial flow, they appear as spirals. The angle of inclination of spirals in relation to flow direction increases with an increase in rotation speed, and at significant dominance of rotational flow spiral clots of vortices can possibly degenerate into toroidal ones. Such a phenomenon has already been observed, however, for laminar flows at Re $\sim$ 300 and rotation $N > 3$ when the phenomenon is observed for laminar regimes when the vortices of the Görtler-type degenerated into Taylor vortices with increasing rotation [33].

The distribution of axial friction at the inner and outer cylinder is shown in Figure 9b,c, respectively. Strips of inhomogeneity in the wall friction are formed by vortices. The increase in vortex angle and decrease in the vortices' size with increasing rotation $N$ are reflected by inhomogeneities in the wall friction. At rotation $N = 4$, the vortices are deviated very far from the channel axis so that the axial friction can become negative; such areas are marked in blue in Figure 9b.

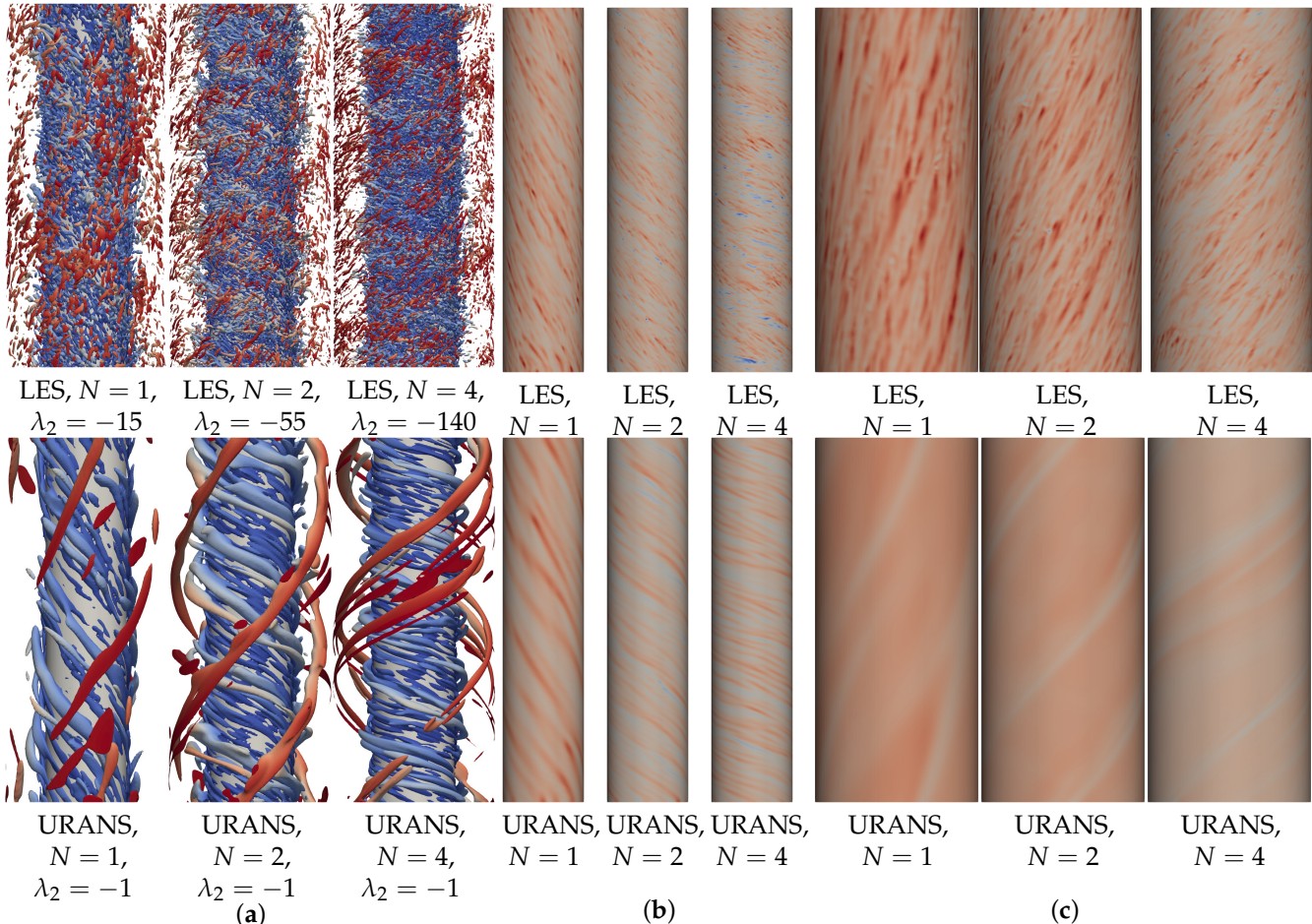

**Figure 9.** Flow structure visualization by LES and URANS. (**a**) Vortices visualisation by $\lambda_2$ isosurfases. Coloured by radius, thus, blue corresponds to the inner cylinder and red corresponds to the outer one. (**b**,**c**) Axial shear stress at the inner and outer cylinders, respectively. Red is positive, blue is negative friction. The same scale is applied for Figures (**b**,**c**).

LES resolves significantly smaller vorticity structures than URANS, thus, URANS visualizations look like LES flow but filtered at a larger scale (Figure 9a). Groups of small vortices united into one larger and longer vortex. This is clearly visible in cases with $N = 2$ and $N = 4$, where clusters of small vortices at the outer wall (red-colored) obtained by LES transformed into larger and longer continuous spiral vortices. The axial shear stress distribution obtained with URANS saves the slope angles of the inhomogeneities formed by the vortex structures, but the distribution is greatly smoothed due to the approach itself (Figure 9b,c). Overall, it should be noted that URANS correctly shows the structure of the flow, albeit in an enlarged form and not in as much detail, but it saves computational resources considerably.

### 3.3. Integral Parameters

In this paragraph, we will consider the integral parameters, axial friction, responsible for pressure loss and tangential friction, responsible for the torque acting on the inner and

outer cylinders. Values of axial friction on the inner and outer cylinder are shown in Table 3. The results of LES show that without rotation, the friction on the inner and outer cylinder is comparable. As the rotational velocity increases, the friction on the inner cylinder grows faster because vortices are more intensely formed on it, which increases energy dissipation. Also, at rotation $N < 0.5$, friction on both walls increases non-linearly with rotation, while at higher values of rotation the dependence is practically linear (Table 3). We have seen a rearrangement of the rotational velocity profile (Figure 1b) and the behaviour of structural parameter $a_1$ (Figure 7a) changes at $N > 0.5$. It changes the momentum transfer mechanism and friction dependence in rotation. Also, this is true for the total pressure loss (Figure 10a).

**Table 3.** Skin friction factor at the inner ($C_{f,i}$) and outer ($C_{f,o}$) cylinder. The difference with LES is written in brackets.

|          | *N* | LES | URANS | EBM | *k-ω* SST |
|----------|-----|-----|-------|-----|-----------|
|          | 0   | 0.00864 | 0.00921 (+6.6%) | 0.00903 (+4.5%) | 0.00921 (+6.6%) |
|          | 0.2 | 0.00946 | 0.00925 (−2.2%) | 0.00903 (−4.6%) | 0.00925 (−2.2%) |
| $C_{f,i}$ | 0.5 | 0.01214 | 0.01192 (−1.8%) | 0.01074 (−12%) | 0.00969 (−20%) |
|          | 1   | 0.01659 | 0.01537 (−7.4%) | 0.01423 (−14%) | 0.01065 (−36%) |
|          | 2   | 0.02311 | 0.02122 (−8.2%) | 0.02138 (−7.5%) | 0.01364 (−41%) |
|          | 4   | 0.03493 | 0.02940 (−16%) | 0.03385 (−3.1%) | 0.01712 (−51%) |
|          | 0   | 0.00754 | 0.00819 (+8.6%) | 0.00810 (+7.2%) | 0.00819 (+8.6%) |
|          | 0.2 | 0.00778 | 0.00823 (+5.8%) | 0.00810 (+4.1%) | 0.00823 (+5.8%) |
| $C_{f,o}$ | 0.5 | 0.00853 | 0.00820 (−3.9%) | 0.00834 (−2.2%) | 0.00842 (−1.2%) |
|          | 1   | 0.00984 | 0.00899 (−8.7%) | 0.00935 (−4.9%) | 0.00866 (−12%) |
|          | 2   | 0.01294 | 0.01129 (−13%) | 0.01256 (−2.9%) | 0.00992 (−23%) |
|          | 4   | 0.01979 | 0.01521 (−23%) | 0.01989 (0.5%) | 0.01169 (−41%) |

Most authors disregard the analysis of the tangential friction or the torque acting on the cylinders, although this may be an important parameter both from a fundamental point of view for tuning turbulence models, and from a technical point of view for solving engineering problems. Here we show non-dimension torque $T$ nondimensionalized using analytical solutions that are proportional to viscosity and rotation. With good accuracy, it can be assumed that the dimensionless torque is proportional to rotation at $N > 0.5$, then, its dimensional value is proportional to the square of the rotation frequency of the inner cylinder (Figure 10b).

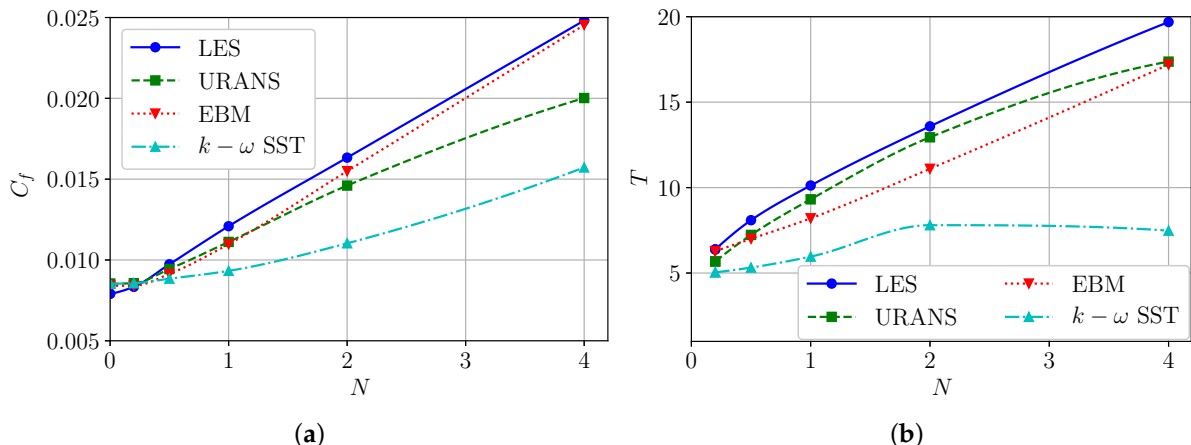

**Figure 10.** Dependence of the skin friction coefficient (**a**) and torque acting on cylinders (**b**) in rotation.

As has been shown many times, models with the eddy viscosity hypothesis are unable to adequately capture swirling flows [13]. Thus, the *k-ω* SST model underestimates friction at the inner cylinder more where more vortices are formed (Table 3). At rotations, $N < 0.5$ gives an acceptable error in pressure loss (8–9%), but with increasing rotation,

the underestimation grows to 37%. Average difference in pressure loss is 26%. The torque is underestimated by this approach from 20% up to 60%, with an average value of 46%. Thus, tangential friction is much worse predicted in a steady-state simulation by the $k$-$\omega$ SST model.

The disadvantages of the stationary $k$-$\omega$ SST approach are partially compensated for by the non-stationary URANS approach, which is capable of resolving non-stationary vortex structures [16,34,35]. The behaviour of pressure losses by URANS at a low rotation ($N \leq 0.2$) is similar to RANS, clearly seen in Figure 10a and Table 3. Thus, URANS at small rotations is not sensible to rotation. This is happening because high eddy viscosity suppresses the formation of instabilities and vortices. As rotation increases, the difference in URANS with LES in pressure loss increases, but not exceeding 20%, with an average value of 11%. The torque is predicted by URANS quite well, with maximum difference less than 12% and an average value around 8% (Figure 10b).

The EBM reproduces Reynolds stresses well (Figure 4) and thus proved to be suitable for describing swirling flows. It is interesting to note that the EBM approach overestimates the axial friction at the axial flow on the channel walls, as does the $k$-$\omega$ SST approach, but as the rotation increases, the difference first increases and then almost disappears. The maximum difference with LES for wall friction is 14% and for total pressure loss 9% (average 5%). The torque is predicted satisfactorily, with an error of up to 19% (average 16%).

## 4. Conclusions

LES simulations of a fully-developed turbulent annular flow at Re $= 10,000$ and $r_o/r_i = 0.5$ with an axially rotating inner wall at a range of rotation rates $N = U_\theta/U_b = 0$, 0.2, 0.5, 1.0, 2.0, and 4 was carried out to study the effects of rotation on the flow, turbulence properties, and eddy structures. The observation and analysis of the mean velocity, turbulent stress fields, the vortical structures visualized by the $\lambda_2$, and integral parameters, as well as comparisons made between them and the results of URANS $k$-$\omega$ SST, RANS $k$-$\omega$ SST, and EBM, lead to the following conclusions:

- Rotation leads to a thinning of the viscous sublayer and, as a consequence, a widening of the mean axial profile and an increase in gradients at the wall. The axial velocity component distribution in wall units decreases due to an increase in friction on the wall with increasing rotation $N$.
- Rotation decreases axial fluctuation production and increases tangential fluctuation production in wall units, while the maximum value of total production is weakly dependent on rotation, but its position shifts towards the wall into the buffer zone.
- With increase in rotation and $N > 0.5$, the following changes are observed:
  - The tangential velocity component changes its shape, eliminating the local maximum at about one-third of the channel width from the inner wall.
  - The profile of the Reynolds stress tensor component $\overline{u_\theta u_z}$ changes its monotonicity and becomes positive in areas where it was not before.
  - The structural parameter $a_1$ begins to decrease with increasing rotation, i.e., the production of shear components becomes less efficient.
- The applicability of URANS and EBM techniques for describing first and second statistical moments of velocity fluctuation as well as integral characteristics of the flow is shown. URANS describes the vortex structures of the flow well, however, in an enlarged form.

As already written in the introduction, flow in an annular channel is encountered in many applications such as chemical reactors, heat exchangers, turbomachines, and drilling. The fluids used for drilling have non-Newtonian rheology. The turbulent flow of such fluids is not widely studied. Thus, the turbulent flow of non-Newtonian fluids in an annular channel is both interesting from a fundamental point of view and has important applications. Research of this type should be a logical continuation of this study.

**Author Contributions:** Conceptualization, A.G. and Y.I.; methodology, A.G. and Y.I.; validation, A.G.; formal analysis, Y.I.; investigation, A.G. and Y.I.; resources, Y.I.; data curation, Y.I.; writing—original draft preparation, Y.I.; writing—review and editing, A.G. and Y.I.; visualization, Y.I.; supervision, A.G.; project administration, Y.I.; funding acquisition, A.G. and Y.I. All authors have read and agreed to the published version of the manuscript.

**Funding:** The research has been funded by Baker Hughes and by the Russian Science Foundation Grant No. 23-79-30022.

**Data Availability Statement:** Data from the present research effort are available upon request.

**Acknowledgments:** Authors would like to thank the Baker Hughes company for the permission of publishing the research results.

**Conflicts of Interest:** The authors declare no conflict of interest.

## Nomenclature

| | |
|---|---|
| $a_1$ | structure parameter = $\sqrt{\overline{u_z u_r}^2 + \overline{u_r u_\theta}^2 + \overline{u_\theta u_z}^2}/2k$ |
| $C_f$ | skin friction factor = $\tau_z/0.5\rho U_b^2$ |
| $D_h$ | hydraulic diameter = $2(r_o - r_i)$ |
| $k$ | turbulent kinetic energy = $0.5(\overline{u_r u_r} + \overline{u_\theta u_\theta} + \overline{u_z u_z})$ |
| $L_z$ | computational length in the $z$ direction |
| $N$ | rotation rate = $U_\omega/U_b$ |
| $N_r, N_\theta, N_z$ | number of mesh points in the $r, \theta, z$ directions, respectively |
| $\mathbf{P}$ | production of Reynolds stress tensor |
| $r_i, r_o$ | radius of inner and outer cylinder, respectively |
| Re | Reynolds number based on characteristic velocity and length scales |
| $\mathbf{S}$ | symmetric component of the velocity gradient tensor |
| $U_r, U_\theta, U_z$ | mean velocity components in the $r, \theta, z$ directions, respectively |
| $U_b$ | bulk axial velocity |
| $U_\omega$ | inner cylinder rotation velocity |
| $U_\tau$ | friction velocity = $\left(\sqrt{\tau_z^2 + \tau_\theta^2}/\rho\right)^{1/2}$ |
| $u_r, u_\theta, u_z$ | fluctuating velocity components in the $r, \theta, z$ directions, respectively |
| $T$ | dimensionless torque = $\frac{\tau_{\theta,i}(r_o^2 - r_i^2)r_i}{2r_o^2 \nu \rho U_\omega}$ |
| $y$ | distance from the inner or outer wall |
| $\Delta r, \Delta \theta, \Delta z$ | mesh spacing in the $r, \theta, z$ directions, respectively |
| $\lambda_2$ | second largest eigenvalue of $\mathbf{S}^2 + \mathbf{\Omega}^2$ |
| $\mathbf{\Omega}$ | anti-symmetric component of the velocity gradient tensor |
| $\rho$ | density of fluid |
| $\tau_\theta, \tau_z$ | statistically averaged wall shear stress in the $\theta, z$ directions, respectively at the inner or outer wall |
| $()^+$ | value in wall units, normalized by $U_\tau, \nu$ |
| $()_{,i}, ()_{,o}$ | value related to inner or outer wall, respectively |
| $()_{rms}$ | root mean square value |

## Abbreviations

| | |
|---|---|
| CFL | Courant–Friedrichs–Lewy number |
| DNS | Direct Numerical Simulation |
| LES | Large Eddy Simulation |
| URANS | Unstationary Reynolds Average Navier–Stokes |
| RANS | Reynolds Average Navier–Stokes |
| RSM | Reynolds Stress Model |
| r.m.s. | root mean square |
| EBM | Elliptic Blending Model |
| QUICK | Quadratic Upstream Interpolation for Convective Kinematics |
| TKE | Turbulence Kinetic Energy ($k$) |

### Appendix A

In this appendix we want to bring the discussion of the applicability of the chosen mesh for the LES and URANS methods, since this is a side but important task during simulation. As already mentioned in the main text of the article, from the walls of the computational domain a sparse grid is made with a growth factor of no more than 5% towards the middle of the channel gap. The choice of such a thickening factor is not optimal from the point of view of computational resource consumption, because the mesh is excessively detailed in the radial direction for LES and URANS. While it is recommended not to exceed the proportion of the modelled part of the TKE of 20% for LES [29], for all the selected meshes it is on average not more than 6%, but does not exceed 9% (Figure A1). Since for each rotation $N$ the mesh varied but the construction algorithm was the same, we will make an analysis using the mesh parameters and time step for $N = 1$ as an example. The obtained conclusions and estimations can be extended to other rotations $N$ as well, since the sizes of the near-wall cells in viscous scales and CFL numbers are of approximately the same order for all rotations.

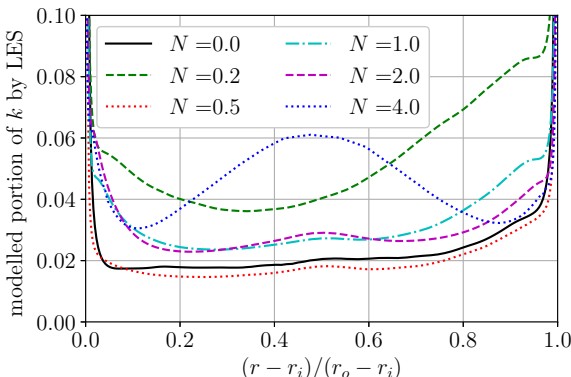

**Figure A1.** Portion of modelled $k$ (TKE) by LES

A Richardson extrapolation is usually used to estimate the error of the numerical algorithm [36]. Extrapolation of the solution is performed on a series of grids and time steps. For a Richardson extrapolation to provide reliable estimates of the exact solution of the model, several basic assumptions should be satisfied: (1) that the discrete solutions are within the asymptotic range, (2) that the meshes have uniform (Cartesian) spacing over the domain, and (3) that the coarse and fine meshes are connected by systematic refinement. While it is possible to reach obtaining the discretisation parameters in the asymptotic range, the uniformity of the grid and the series of the grid's systematic refinement are rather difficult to implement for problems of such scales. Therefore, we should specify at once that the obtained errors for this study are conditional and approximate.

It is also important to note that this procedure is not fully applicable to LES due to the direct dependence of the dynamic model of the subgrid viscosity on the cell size. It is not quite clear how proportions between resolved fluctuation and modelled ones in URANS depends on discretization. The LES method converges with DNS with grid refinement and time step, which is not the case for URANS.

The basic parameters of the grid and time step will be the ones given for $N = 1$ in Table 1 for LES and in Table 2 for URANS. We denote the time step by $\Delta t$ and the average cell size by $\Delta h = \sqrt[3]{\Delta V}$, where $\Delta V$ is the average cell volume. For the basic simulation, we denote the corresponding values by $\Delta t_0$ and $\Delta h_0$.

Tables A1 and A2 show the results of simulations with different grid and time step parameters. The standard Richardson extrapolation is defined in the below formula:

**Table A1.** Dependence of pressure loss coefficient $C_f$ and torque coefficient $T$ on mesh and time step resolution for $N = 1$ obtained by LES.

| Discretization Parameters | | $C_f$ | | | $T$ | | |
|---|---|---|---|---|---|---|---|
| $\Delta t/\Delta t_0$ | $\Delta h/\Delta h_0$ | Value | $Err_{Rich}$, % | GCI, %, $\Delta t \mid \Delta h$ | Value | $Err_{Rich}$, % | GCI, %, $\Delta t \mid \Delta h$ |
| 1 | 1 | 0.0120902 | 2.0 | 3.6 \| 1.3 | 10.12200 | − 8.6 | 3.5 \| 6.2 |
| 1 | 0.666 | 0.0121562 | 2.6 | − \| − | 10.39252 | −6.1 | − \| − |
| 1 | 1.5 | 0.0114605 | −3.3 | − \| 4.2 | 9.68278 | −12.5 | − \| 5.0 |
| 0.666 | 1 | 0.0119124 | 0.5 | − \| − | 9.97591 | −9.9 | − \| − |
| 0.83 | 1 | 0.0119603 | 0.9 | 2.2 \| − | 10.02131 | −9.5 | 2.5 \| − |
| 0 | 0 | 0.0118497 | 0 | − | 11.06893 | 0 | − |

**Table A2.** Dependence of pressure loss coefficient $C_f$ and torque coefficient $T$ on mesh and time step resolution for $N = 1$ obtained by URANS.

| Discretization Parameters | | $C_f$ | | | $T$ | | |
|---|---|---|---|---|---|---|---|
| $\Delta t/\Delta t_0$ | $\Delta h/\Delta h_0$ | Value | $Err_{Rich}$, % | GCI, %, $\Delta t \mid \Delta h$ | Value | $Err_{Rich}$, % | GCI, %, $\Delta t \mid \Delta h$ |
| 1 | 1 | 0.011135 | −8.3 | 1.0 \| 6.8 | 9.25550 | −17.0 | 0.9 \| 11.5 |
| 1 | 0.666 | 0.011460 | −5.6 | − \| − | 9.72499 | −12.8 | − \| − |
| 1 | 0.8 | 0.011317 | −6.8 | − \| 8.5 | 9.40816 | −15.6 | − \| 22.1 |
| 0.5 | 1 | 0.011028 | −9.1 | − \| − | 9.17314 | −17.7 | − \| − |
| 0.75 | 1 | 0.011092 | −8.6 | 1.4 \| − | 9.20806 | −17.4 | 0.9 \| − |
| 0 | 0 | 0.012137 | 0 | − | 11.15210 | 0 | − |

$$f(\Delta h, \Delta t) = f_0 + a\Delta h + b\Delta h^2 + c\Delta t + c\Delta t^2, \tag{A1}$$

where $f$ is the numerical solution, $f_0$ is the extrapolated solution, and $a$, $b$, $c$, $d$ are the coefficients of the series. The parameters $f_0$, $a$, $b$, $c$, and $d$ can be found by solving an optimization problem on the set of value $f$ at different parameters, $\Delta h$ and $\Delta t$. This extrapolation was applied, which gave extrapolated values of $C_f$ and $T$. These values are included in Tables A1 and A2 for $\Delta t = 0$ and $\Delta h = 0$. The error of $f(\Delta h, \Delta t)$ with respect to the relatively extrapolated value $f_0$ is determined as follows:

$$Err_{Rich}(\Delta h, \Delta t) = \frac{(f(\Delta h, \Delta t) - f_0)}{f_0} \cdot 100\%. \tag{A2}$$

and included in Tables A1 and A2. The error estimate of the LES method for the pressure loss coefficient $C_f$ is 2% and 8.6% for the torque coefficient $T$. For the URANS method, the corresponding error estimations are 8.3% and 17%, respectively.

The Roache's grid convergence index (GCI) is also evaluated, assuming second order of convergence in space and time [36]. The GCI for the fine grid numerical solution is defined as:

$$GCI = \frac{F_s}{r^p - 1}\left|\frac{f_2 - f_1}{f_1}\right| \cdot 100\%, \tag{A3}$$

where $F_s = 3$ is a factor of safety, $p = 2$ order of accuracy, $f_1$ is value obtained using fine time or space resolution, and $f_2$ is value obtained using coarser time or space resolution. The calculation of the convergence index $GCI$ requires a solution on two grids or with two different time steps and the order of discretisation accuracy. The estimates given in Tables A1 and A2 are made with respect to the smallest time step and space resolution. Since the $GCI$ convergence index is given separately for space resolution and time resolution, we will take the larger one for estimation. The convergence index $GCI$ of the LES method for the pressure loss is 3.6% and for the torque it is 6.2%. For the URANS method, the corresponding estimates are 6.8% and 11.5%, respectively.

　　　　　LES for the selected parameters showed a small error in determining the pressure losses andan acceptable error in determining torque. With a sufficiently detailed mesh, applicable even for the unresolved LES, URANS shows a rather high error. Apparently, an excessively detailed mesh has a negative effect on the URANS solution. The problem of grid convergence of the URANS method in solving problems with centrifugal force requires a special study, which is beyond the scope of this paper. Thus, the choice of grid and time step can be considered reasonable for LES and satisfactory for URANS.

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
