# Peer review of "Numerical Simulation of Taylor—Couette—Poiseuille Flow at Re = 10,000"

_fluids, doi:10.3390/fluids8100280_

Round 1

Reviewer 1 Report

LES simulations of a fully-developed turbulent annular flow at Re = 10, 000 and Ro/Ri = 0.5.

The paper is well written and the presented results is well explained by the authors.

The future scope and  scientific application of the work need to be added in the conclusion section.

Author Response

Dear reviewer thank you for your time and fast review! Below you can find your comments (C) and our answers (A).

  1. C: The future scope and scientific application of the work need to be added in the conclusion section.

A: In the introduction, the applied tasks of the research were formulated. In the conclusion the information will be duplicated, information on further directions of work is added. (line 367)

  1. C: Reason for the study is not convincing. I did not find anything very new in their findings. The scientific contribution of this manuscript is limited however, the comparison of various numerical scheme is interesting. Therefore, I am in favour of publication.

A: Previous studies for this scale of Reynolds number were limited to a rotation of N<1 (information has been added at line 65). Here we extended it to N=4 (the first goal). The results are quite expected. In this case we have shown that there is no big difference compared to smaller rotations. Non-significant changes in the behavior of some Reynolds stresses were found. Pressure and torque losses are presented, which may be interesting for industrial applications. A comparison with less time-consuming approaches (second objective) is made, which is also very important in engineering approaches.

  1. C: “URANS and EBM approaches show good agreement with LES in mean flow, turbulent statistics, and integral parameters.” What is the percentage deviation in the approach?

A: There are a lot of parameters to compare. We added the pressure loss as the most interesting in the abstract.

  1. C: Schematic diagram of the numerical scheme need to be incorporated in text.

A: Numerical schemes are well described in the text and there are standard for LES and other applied methods. Adding an additional table or diagram to an already existing text description only increases the size of the article without adding new information. We don’t see reasons to do that.

  1. C: Near the wall and away from the wall “discussion on meshing is required”.

A: Information about grid condensation is added to the text.

  1. C: The mesh y+ properties of the relevant study must be specified. The grid resolution of the simulations as well as grid independent test is required.

A: For LES and URANS mesh resolution and mesh size at the wall in viscous units are presented in Tables 1 and Table 2 respectively. Usage of is generally agreed for LES and twice larger than. LES has sub-grid dissipation model that depends on grid size. Mesh convergence is not so applicable to LES method. There are some recommendations in the literature about meshing for LES. In the current study, mesh was chosen based on these recommendations and relative studies.  Some investigation of mesh size and time step added in appendix. Appropriate information is added to text.

  1. C: In Fig. 5 and Fig. 10, legend is overlapped with the figure that need to be adjusted accordingly.

A: Fixed

Reviewer 2 Report

Overall a very nice work, and I have also learned a lot from this nice paper. several comments from my side:

1) it would be better to give the governing equation for incompressible fluid before Eq.1) and 2) are given. Whether it is in a conserved form or not?

2) CFL < 0.9 is used. Usually we select CFL < 0.5. Any reason ?

3) it would be nice, if authors provide the model parameter values used in the subgrid model. This may help to re-produce the results.

Author Response

Dear reviewer thank you for your time and fast review! Below you can find your comments (C) and our answers (A).

  1. C: it would be better to give the governing equation for incompressible fluid before Eq.1) and 2) are given. Whether it is in a conserved form or not?

A: We see no reason to write out again the system of Navier-Stokes equations for an incompressible Newtonian fluid. We do not add new terms to the equation. The equations are generally known.

  1. C: CFL < 0.9 is used. Usually we select CFL < 0.5. Any reason ?

A: An unconditionally stable Crank-Nicholson scheme of the second order of accuracy is used. There is no need to observe any conditions on the CFL. There are no noticeable changes in the mean distributions when reducing the time step to CFL < 0.5. Analysis of the change in integral parameters with decreasing time step is added to the appendix. We consider the selected parameters to be satisfactory.

  1. C: it would be nice, if authors provide the model parameter values used in the subgrid model. This may help to re-produce the results.

A: The dynamic subgrid model was used without changes of any parameters as it was described in referred articles in references.